# Study of risk perception consumption behavior of sports tourism in China

**Gang Li**[1☯], **Yan Cheng**[2☯], **Jie Cai**[1☯]*

1 School of Sports, Shandong University of Finance and Economics, Jinan, Shandong, China, 2 Shandong Analysis and Test Center, Qilu University of Technology (Shandong Academy of Sciences), Jinan, Shandong, China

☯ These authors contributed equally to this work.
* 314765618@qq.com

## Abstract

In order to further implement the "Healthy China 2030 Planning Outline" and actively develop the sports tourism industry, this study analyzed the independent variables and dependent variables that affect the urban residents' sports tourism risk perception consumption behavior.Finally, the simultaneous regression, stepwise regression and hierarchical regression models of sports tourism risk perception consumption behavior of urban residents are constructed. The purpose is to help people effectively screen all kinds of risks in the process of sports tourism, promote the growth of sports consumption, and provide theoretical and practical guidance for the development of sports tourism

## 1. Introduction

Since the mid-1990s, with the gradual development and expansion of the global and popular sports tourism consumption market represented by fitness entertainment and leisure tourism, sports tourism has become an important part of modern people's life. Although the sports tourism market has huge potential at present and presents a momentum of rapid development, sports tourism products still occupy a small proportion in the whole tourism market. The reason is not that sports tourism products are not attractive enough, but that sports tourism products themselves contain too many risk factors. Tourists will face risks when they participate in tourism. This will directly improve the level of risk perception of sports tourism consumers, so that they will stay away from many sports tourism products and choose tourist tourism products with low relative risk coefficient. How to accurately grasp the influencing factors of people's risk perception consumption behavior in the process of sports tourism, reasonably construct the risk perception consumption behavior model of sports tourism, and understand the risk perception consumption behavior of consumers will become an important problem to be solved urgently. This study enriches sports tourism theory and risk perception theory, comprehensively grasps all kinds of risks faced by people in the process of tourism. By comparing the changes in risk perception of different groups, this study carried out corresponding market segmentation, formulated corresponding marketing strategies and optimized product design,

**Data Availability Statement:** Data are available from the Ethics Committee of School of Sports, Shandong University of Finance and Economics (contact via 0531-82911131) for researchers who meet the criteria for access to confidential data.

**Funding:** The authors received no specific funding for this work.

**Competing interests:** The authors have declared that no competing interests exist.

so as to meet the different needs of people and promote the development of sports tourism market in China.

## 1.1 Literature review

**1.1.1 Urban sports tourism.** With the rise and development of sports tourism, sports tourism research has received extensive attention among many scholars such as sports science, tourism, and economics, which has also aroused the interest of domestic and foreign researchers in sports tourism research. De Knop [1] pointed out that the interrelationship between sport and tourism in five European countries and discussed in detail the basic requirements and precautions for conducting sports tourism research, this lays a foundation for the future sports tourism research. The relevant situation should be fully considered when researching, and the potential tourism components of tourists' participation in potential sports activities and participation activities should be fully recognized, but the question of the risks of sports tourism was not raised [2]. Tourism is an increasingly salient global economic sector, mountain destinations and the multi-billion dollar ski industry are considered particularly vulnerable to climate variability and change, the research finds that there are economic risks, social cognitive risks and knowledge risks in tourism, this is a very significant finding [3]. Each motivation of golfers and their intention to return to the golf course has differed on their satisfaction with the services or product that provide of the golf course and this makes the golf management must need to know the answers to maximize their profits and continue to growth, the study highlights the importance of tourism products and services [4]. The practice of sports activities related to nature and the environment is a social and tourist phenomenon that is increasingly popular, they find and analyze the sociodemographic characteristics of tourists who take trips in order to find and enjoy trekking opportunities, this study emphasized demographic characteristics but did not address issues such as the risks people may face while traveling [5]. The convergence of sports and tourism industries is a vital direction for the coordinated development of industries, and a vital means to build a quality life circle suitable for living, working and traveling in the urban agglomeration of the Guangdong–Hong Kong–Macao Greater Bay Area [6]. Perić(2010)accentuated the connection between sport and tourism and to analyze the motives and profiles of participants in sports events.sport in the modern tourism has not only a perceptual role, but it is also important contents of stay, and frequently the main motive for travelling to a certain tourist destination, this study highlights the importance of sports tourism [7]. Gozalova et al.(2014) analyzed sports tourism in Russia, concluded that there exists a high correlation dependence of foreign and domestic development of sports tourism on resources allocated for sports infrastructure; and sports tourism tours draw visitors to their favorite sporting event, facility, or destination throughout the world [8]. Rynik and Gibas (2019) used the spatial dispersion indicator of residential buildings in relation to sports infrastructure buildings to identify urban sports zones (also known as sports cities) as a potential product of urban or sports tourism in Poland The findings can be used to identify areas that could serve as sports cities, which are defined as varied, separate, large areas, whose development is associated with sport and recreation [9]. F Zhang used the method of big data analysis to carry out the statistical analysis of Eco-sports Tourism behavior in urban space, to guide the management of urban planning and sustainable development of sports tourism, and puts forward the modeling and analysis method of Eco-sports Tourism behavior based on data mining technology [10]. B Dai took the international marathon of Bayuquan as an example,to explore the influence of large-scale sports events to urban tourism,the results shows:the international marathon can improve the popularity of Bayuquan,accelerate the construction of urban facilities and environment,and promote the cultural exchange of urban tourism [11].

Song and Wang, through the literature method, comparative method, carefully analysed some cities and found that the development of sports culture played an important role in promoting the development of tourism. But at the same time we also need to eliminate some negative effects. Only when we seize the opportunity to create favorable conditions to accelerate the development, improve the reputation of the city, and create a healthy sports culture tourism brand, can we enhance the competitiveness of the city [12]. González-García investigated residents' perceptions of the economic impact of sports tourism according to the population of residence, differentiating between rural, urban and coastal populations. The results of this study indicate that residents have a high regard for the impacts associated with sports tourism and that perceptions do not vary greatly, depending on the population of residence of the respondents, as well as age and sex [13]. Chang and Pei used literature method, interview method, field survey method of "big Yiyang city circle" the sports tourism development strengths, weaknesses, opportunities and threats for in-depth analysis, and put forward unified planning and dislocation development. Highlight the features, building brands; information sharing, marketing collation; to improve facilities and to attract investment and to improve the concept of service development [14]. Thus, On the basis of sports tourism, resident urban sports tourists are urban residents temporarily leave their own way of life and place to participate in or experience sports activities, nature interest, nostalgic tourism activities, leisure, entertainment, fitness, adventure activities.

## 1.2 Risk perception

Lawonk [15] examined the multiple dimensional aspects of perceived value in relation to purchase intention in the Eco-tourist industry. Additionally, the study evaluated the influence of trust and perceived risk as mediators on perceived value, and thus, on purchase intention. Findings showed that emotional value, functional value, boredom alleviation value, epistemic value and perceived risk are significant predictors of influenced Eco-tourism purchase intention, this study provides important implications for the study of sports tourism risk [16]. Tourism has many impacts on residents' quality of life. One outcome that can result from these impacts is the experience of stress. Individuals who experience stress are at increased risk for mental and physical illnesses. This paper examines the relationship between residents' perceived impacts of tourism on their quality of life and their experience of tourism related stress. A logistic regression model revealed that perceived risk was significantly related to the experience of tourism related stress, risks such as responsibility, knowledge, function and psychological risks may exist in the process of sports tourism. Tourism decision makers can address tourism quality of life impacts in these specific areas to help mitigate tourism related stress. Gurtner found that [17] since the turn of the century the global tourism industry has been impacted by an increasing and diverse range of external shocks. As an industry that generates hundreds of billions of dollars annually through income revenues, employment, investment and infrastructure, there has been a growing interest in trying to develop more effective management strategies to prevent and/or mitigate the adverse effects of such events, particularly at the localised region or destination level. Ritchie et al. [18] believes that there is a close relationship between tourism risk, crisis and disaster management, and future research should pay attention to the risks in tourism to improve people's willingness to travel, the interpretation of tourism risk is of great significance to this study. Williams et al. (2013) figured out that age, and tolerance of both general and tourism-specific risks, were associated with the importance of hazards as deterrents to tourist behaviour, men with lower education background are more willing to take risks than women with higher education background, conclusions on age and educational background are used in this paper [19]. Cui F et al. (2016) systematically reviewed

existing researches of tourism risk perception,pointed out that there is a certain critical value for travel risk perception of tourists; Cognitive ability is an important factor affecting the level of tourists objective risk perception; and quantitative assessment of tourism risk perception level is helpful to the tourism decision making and destination management, the suggestion of cognitive ability is of reference significance to the risk perception in this study [20]. Tourist destination risk perception exerted a negative effect on tourism image, culture image, and stability image of the destination; the risk perception of tourists on tourism destinations would lead to a change in tourism attitudes, and the increased risk perception of tourists on destinations had negative effects on behavior intention (Zhang H et al.,2019) [21].

A psychometrically sound instrument—the Construction Worker Risk Perception (CoWoRP) Scale had been developed to assess the risk perception of construction workers. CoWoRP Scale was affirmed to have three dimensions of worker risk perception, namely risk perception—probability, risk perception—severity, risk perception—worry and unsafe. These three dimensions of worker risk perception were negatively correlated with their risk-taking behavior (Man S S.et al.,2019) [22]. A reference model defining various dimensions of occupational risk perception, relevant for the different ethnicities, was identified. The authors identified 4 relevant dimensions: behavioral control, work conditions, safety climate and personal attitude (Ricci F.et al.,2019) [23]. Chaswa et al. (2020) identified six factors as variables that showed a significant effect on workers' perception of risk, include "dreaded factor","avoidability and controllability","expert knowledge","personal knowledge", "education level"and "age" [24]. Many international public-private partnership projects have suffered from frequent project pending status or failure because of dissimilar interests among stakeholders over projects' long development period, time-dependent risk, risk exposure period and risk transfer and responsibility are main risk perception gaps (Park et al., 2020) [25]. The above analysis of risk plays an important guiding significance for the relevant measurement of perceived risk in this study.

According to CoWoRP scale, combined with sport tourism, Chen Xiuping (2019) established the sports tourism consumption risk perception scale to analyze the market role and the relationship among the sports tourism, sports tourism projects and consumers, and in combination with the characteristics of the sports tourism, a kind of sports tourism consumption risk perception scale (STCRPS) that integrates the sports tourism, sports tourism project and consumers is put forward to improve the value in use of the model [26]. Chen [27] established the sports tourism consumption risk perception scale to analyze the market role and the relationship among the sports tourism, sports tourism projects and consumers in the background of the Internet era. In combination with the characteristics of the sports tourism, a kind of sports tourism consumption risk perception scale (hereinafter referred to as the STCRPS for short) that integrates the sports tourism, sports tourism project and consumers is put forward to improve the value in use of the model.Yue [28] put forward the consumption risk model of national sports tourism (referred to as CRMNST for short) based on the integration of the intelligent interactive system model (referred to as IISM for short) to optimize the decision making process of the intelligent interactive system model. The consumption risk model of tourism can be effectively integrated with the sparseness of data being processed, in particular, the offsite consumption risk of national sports tourism. Chen [29] aims to explore the relationship between risk perceptions and destination image and visit intentions with Chinese domestic tourists. He divides the destination image into cognitive and affective dimensions. The current study also examines how three risk perceptions, including physical, financial and performance risks, inuence the intention to visit a destination. Evidence from 336 Chinese domestic tourists who visited Wuhan after COVID-19pandemic suggest that perceived risk negatively and signicantly inuences cognitive and affective images. Shi [30] through the research on the perception,identification and management of sports tourism risks by means of

literature material research method, analyzes the factors and characteristics that affect the perception of sports tourism risks,such as infotainment,severity,voluntary,disaster,variability, controll ability and fear,etc. Meiyang and chenguo investigate the prediction of risk events by regression analysis with dimensions of personality, risk perception and sports, relations between risk events, risk perception, and the facets of the personality dimensions via data collecting from 664 adolescent athletes aged 13–18 years (male 364, female 300) [31]. Kim examined and compared the influence of tourists€ risk perceptions on travel intentions across mega sporting event host destinations with different levels (i.e. apparent risks, less imminent risks, and unidentified risks) of such risks. The results indicated that perceived terrorism risk significantly influenced the tourists€ travel intentions [32]. Through literature review, we believe that the perceived risk of tourism includes seven risks, including economic, information, social cognition, responsibility, knowledge, function and psychological risks.

## 2. Method

According to the literature review, when people prepare to engage in sports tourism consumption, all kinds of risks including economic, information, social cognition, responsibility, knowledge, function and psychological risks naturally enter consumers' behaviors, and their purchase decisions and personality traits will affect consumers' post-purchase behaviors. In fact, purchase decision and personality are the dependent variables and independent variables of sports tourism risk perception consumption behavior, respectively. The study was verbal consent by the Ethics Committee of School of Sports, Shandong University of Finance and Economics(2021/005, May 17, 2021) and was conducted following the Declaration of Helsinki established by the World Medical Association. Submission of the online survey after completion implied consent to participate in this study, this statement has already been written at the beginning of the online questionnaire. The subjects of this study were people aged 18–60 years old who lived in urban areas of China and had fixed occupations. A total of 1000 questionnaires were distributed and 843 valid questionnaires were collected. The effective recovery rate was 84.3%.

## 3.Variable analysis of sports tourism risk perception consumption behavior

### 3.1. Analysis on dependent variables of sports tourism risk perception consumption behavior

**3.1.1. Dimensional identification of purchase decisions.** Sports tourism risk perception included economic (factor 1), information (factor 2), social cognition (factor 3), responsibility (factor 4), knowledge (factor 5), function (factor 6) and psychological (factor 7) risks. As shown in Table 1, the analysis of explained variance shows that only one factor has eigenvalues greater than 1, and the characteristic value of the second factor starts to be less than 1. In addition, the explained variance of the first seven factors is 45.918%, indicating that the dimension of one factor can be identified. In addition, further analysis of the factor load index shows that the whole factor load (shaded part) is between 0.524 and 0.859. This indicates that the factor load of the whole questionnaire has reached the common standard, and the explanatory variables have reached 20%-50% persuasiveness.

**3.1.2. A general analysis of purchasing decisions.** The multiple comparison method in variance analysis is used to analyze the differences in variables such as age, occupation, educational background and personal income.

**Table 1. Eigenvalues and explicable variation.**

| Factor | Original eigenvalue of factor | | | Cumulative load of extracted factor | | |
|---|---|---|---|---|---|---|
| | All eigenvalues | Variation(%) | Cumulative contribution rate (%) | All eigenvalues | Variation (%) | Contribution rate (%) |
| 1 | 3.667 | 52.389 | 52.389 | 3.214 | 45.918 | 45.918 |
| 2 | 0.962 | 13.743 | 66.132 | | | |
| 3 | 0.824 | 11.776 | 77.908 | | | |
| 4 | 0.535 | 7.648 | 85.557 | | | |
| 5 | 0.426 | 6.092 | 91.649 | | | |
| 6 | 0.352 | 5.023 | 96.672 | | | |
| 7 | 0.233 | 3.328 | 100.00 | | | |

Multicollinearity means that the model estimation is distorted or difficult to estimate accurately due to the existence of precise correlation or high correlation between explanatory variables in the linear regression model. Generally speaking, due to the limitation of economic data, the model is improperly designed, resulting in a general correlation between explanatory variables in the design matrix.

This study investigates the relationship between factors such as age, gender, occupation, educational background, income and personality traits and perceived risk consumption behavior, so it is necessary to use multiple comparison method to prevent the model estimation distortion or difficulty to estimate accurately due to the existence of precise correlation or high correlation between dependent variables.

(1) Analysis based on age

Based on the analysis of different ages of sports tourism purchase decisions, as shown in Table 2, multiple comparison difference of Fisher's least significant difference (LSD) test showed that people under the age of < 30 years olds, 30–40 years olds, 41–50 years old and people over the age of 50, there does not appear significant difference. It shows that different age groups in sports tourism have more uniform understanding on consumer purchase decision. There is no differentiation yet.

(2) Analysis based on career

**Table 2. Purchase decision and multiple comparison LSD difference test for different ages.**

| Age | Mean value | Standard error | Mean difference |
|---|---|---|---|
| V1 = age under 30 | 2.863 | 0.6597 | |
| V2 = age between 30–40 | 3.035 | 0.9021 | |
| V3 = age between 41–50 | 3.102 | 0.748 | |
| V4 = age over 50 | 3.107 | 1.1157 | |
| V1-V2 | | | -0.1716 |
| V1-V3 | | | -0.2388 |
| V1-V4 | | | -0.2439 |
| V2-V3 | | | -0.0672 |
| V2-V4 | | | -0.0723 |
| V3-V4 | | | -0.0051 |

*The mean difference is significant at the 0.05 level

**the mean difference is significant at the 0.01 level.

**Table 3. Purchase decision of different careers and multiple comparison LSD difference test.**

| Occupation | Mean value | Standard error | Mean difference |
|---|---|---|---|
| V1 = Organ civil servants | 2.667 | 0.7632 | |
| V2 = Culture, health and sports | 2.571 | 0.5002 | |
| V3 = Posts and telecommunications, electric power and transportation | 3.063 | 0.7261 | |
| V4 = Construction, mining and manufacturing | 3.06 | 0.9888 | |
| V5 = Wholesale, retail, catering and leisure service workers | 2.503 | 0.8267 | |
| V6 = Computer, finance and real estate | 3.446 | 8.834 | |
| V7 = Education, scientific research and technical service workers | 3.026 | 0.6006 | |
| V1-V2 | | | 0.0952 |
| V1-V3 | | | -0.3958* |
| V1-V4 | | | -0.3929* |
| V1-V5 | | | 0.164 |
| V1-V6 | | | -0.7798* |
| V1-V7 | | | -0.3590* |
| V2-V3 | | | -0.4911* |
| V2-V4 | | | -0.4881* |
| V2-V5 | | | 0.0688 |
| V2-V6 | | | -0.8750* |
| V2-V7 | | | -0.4542* |
| V3-V4 | | | -0.003 |
| V3-V5 | | | 0.5599* |
| V3-V6 | | | -0.3839* |
| V3-V7 | | | 0.0369 |
| V4-V5 | | | 0.5569* |
| V4-V6 | | | -0.3869* |
| V4-V7 | | | 0.0339 |
| V5-V6 | | | -0.94338* |
| V5-V7 | | | -0.5230* |
| V6-V7 | | | 0.4208* |

*The mean difference is significant at the 0.05 level

**the mean difference is significant at the 0.01 level.

The analysis of sports tourism purchase decision based on different occupations is shown in Table 3. Further multiple comparison of LSD difference test shows that the group of civil servants and posts and telecommunications, electricity, transportation; Construction, mining and manufacturing; Computer, finance and real estate; There were significant differences in the dimension of purchasing decision among the four types of occupation, education, scientific research and technical service workers ($P < 0.05$).

There was significant difference in the dimension of purchase decision-making between the two groups that posts and telecommunications, electric power, transportation groups and wholesale, retail, catering and leisure service workers; Computer, finance and real estate ($P < 0.05$).

There was significant difference in the dimension of purchase decision-making between the two groups that construction, mining and manufacturing groups and wholesale, retail, catering and leisure service workers; Computer, finance and real estate ($P < 0.05$).

There were significant differences in purchase decision-making between computer, finance and real estate groups and education, scientific research and technical service workers (P < 0.05).

To sum up, we believe that different occupational groups have different purchasing decisions. Occupation may influence purchasing decisions.

(3) Analysis based on educational backgrounds

The analysis of sports tourism consumption purchase decision based on educational backgrounds is shown in Table 4. Further multiple comparison LSD difference test shows that compared with the people of the high school and below, the people with undergraduate and graduate education is more willing to buy relevant sports tourism products, and there is a significant difference between the two, P < 0.05. Compared with the people with college and undergraduate education, the people with graduate education are more willing to buy relevant sports tourism products, and there is a significant difference between them (P < 0.05). This shows that the higher the educational level, the stronger the decision to buy sports tourism products. Educational level may have an impact on purchasing decisions.

(4) Analysis based on individual income

The analysis of sports tourism consumption purchase decision based on different individual income is shown in Table 5. Further multiple comparison LSD difference test shows that compared with the group whose individual income less than 3000 yuan, the group whose individual income above 7001 yuan shows higher purchase decision, and there is a significant difference between them (P < 0.05); Compared with the group whose individual income is between 3001–5000 yuan, the group whose individual income is more than 7001 yuan showed higher purchase decision-making, and there was a significant difference between them (P < 0.05). This shows that the higher the income level, the more willing they are to buy relevant sports tourism products and the stronger the purchase decision. Income level may have an impact on purchasing decisions.

## 3.2. Independent variable analysis of sports tourism risk perception and consumption behavior

**3.2.1. Dimension identification of personality traits.** The BFI-Fr is an effective tool to assess the five dimensions of the five-factor model of personality [33]. The psychometric qualities are excellent in several validation studies, This study adopts the research structure of the scale to classify personality traits into kindness, emotional sensitivity, experience openness, diligence and integrity and extroversion. Use the principal component analysis method to extract the factors that contribute the most to the commonality of each variable, and select the appropriate number of factors according to the extraction standard with the eigenvalue greater than 1. Then, use the maximum orthogonal rotation method to maximize the variation of each common factor in the factor load table, so as to name each group of factors and determine each dimension of personality traits. After factor rotation, there were 4 factors with eigenvalues greater than 1, and the cumulative explained variation was 62.15%. The verification results show that after factor analysis, the items of kindness personality and extroversion personality are extracted into the same factor, which is not consistent with the five personality traits to be verified in this study. [2] As shown in Table 6, select the following items according to the research needs.

Therefore, the factor selection is set as five. After orthogonal rotation, the items containing extroversion personality characteristics are independent as the fifth factor. This study verified

**Table 4. Purchase decision of different educational backgrounds and multiple comparison LSD difference test.**

| Educational background | Mean value | Standard error | Mean difference |
|---|---|---|---|
| C1 = high school and below | 2.408 | 0.586 | |
| C2 = college | 2.696 | 0.6892 | |
| C3 = undergraduate | 2.925 | 0.8075 | |
| C4 = graduate | 3.368 | 0.7811 | |
| C1-C2 | | | -0.2875 |
| C1-C3 | | | -0.5169* |
| C1-C4 | | | -0.9598* |
| C2-C3 | | | -0.2294 |
| C2-C4 | | | -0.6724* |
| C3-C4 | | | -0.4430* |

*The mean difference is significant at the 0.05 level

**the mean difference is significant at the 0.01 level.

that after factor extraction, the five factors are consistent with the five personality characteristics. In the mode fitting test, the MSA index is greater than 0.8, and the residual matrix and its partial correlation value are small, indicating that the fitting degree is good and the cumulative interpretation variance reached 67.25%. The five factors are kindness, emotional sensitivity, experience openness, diligence and integrity, extroversion.

**3.2.2. General analysis of personality traits.** Use the multiple comparison method of analysis of variance to analyze the differences in variables such as age, career, educational background and individual income.

(1) Analysis based on age

The personality traits of sports tourism consumers of different ages are statistically analyzed by Analysis of Variance (ANOVA). Levene's test for homogeneity of variance. The Levene's test is mainly used to test the homogeneity of variance between two or more samples. The samples are required to be random and independent. The test is shown in Table 7 below. The overall P values of the variances of the five personality traits are > 0.05. Accept the hypothesis of homogeneity of variance and make corresponding analysis.

**Table 5. Purchase decision of different individual income and multiple comparison LSD difference test.**

| Individual income | Mean value | Standard error | Mean difference |
|---|---|---|---|
| E1< 3000 yuan | 2.871 | 0.5985 | |
| E2 = 3001–5000 yuan | 2.915 | 0.7154 | |
| E3 = 5001–7000 yuan | 3.078 | 0.9233 | |
| E4> 7001 yuan | 3.331 | 1.1492 | |
| E1-E2 | | | -0.0438 |
| E1-E3 | | | -0.2065 |
| E1-E4 | | | -0.4594* |
| E2-E3 | | | -0.1627 |
| E2-E4 | | | -0.4156* |
| E3-E4 | | | -0.2529 |

*The mean difference is significant at the 0.05 level

**the mean difference is significant at the 0.01 level.

**Table 6. Recapitulation statement of personality trait factors.**

| Factor naming | Item | Factor load | Eigenvalues | Cumulative explanatory variation |
|---|---|---|---|---|
| Kind extroversion | Candid | 0.665 | 6.993 | 62.15% |
| | Charitable | 0.623 | | |
| | Active | 0.513 | | |
| | Optimistic | 0.615 | | |
| | Zealous | 0.702 | | |
| | Gregarious | 0.719 | | |
| | 11.Friendly | 0.721 | | |
| | 12.Trustworthy | 0.581 | | |
| Diligence and integrity | 7. Cautious | 0.777 | 2.65 | |
| | 9. Persistent | 0.651 | | |
| | 15.Self demanding | 0.743 | | |
| | 16.Responsible | 0.632 | | |
| Experience openness | 8. Creative | 0.800 | 1.612 | |
| | 10.Imaginative | 0.792 | | |
| | 13.Like thinking | 0.572 | | |
| | 14.Broad interest | 0.652 | | |
| Emotional sensitivity | 17.Melancholy | 0.726 | 1.359 | |
| | 18.Operational | 0.826 | | |
| | 19.Emotional | 0.872 | | |
| | 20.Unsafe | 0.809 | | |

As shown in Table 8, further multiple comparison LSD difference test shows that there are significant differences between people aged under 30 and people aged between 30–40 in five dimensions: kindness, emotional sensitivity, Experience openness, diligence and integrity and extroversion (P < 0.05). Age may have an effect on personality traits.

(2) Analysis based on career

The personality traits of sports tourism consumers of different occupations are statistically analyzed by ANOVA. The homogeneity test of ANOVA is shown in Table 9 below. Except for the dimension of diligence and integrity, the overall P values of the variances of the other four personality traits are > 0.05. Accept the hypothesis of homogeneity of variance and can be analyzed accordingly.

As shown in Table 10, further multiple comparison LSD difference test shows that organ civil servants; Culture, health and sports; Posts and telecommunications, electric power and transportation; Construction, mining and manufacturing; Wholesale and retail catering and

**Table 7. Statistical table of variance homogeneity test of personality traits of different ages.**

| | Levene statistic | Free degree 1 | Free degree 2 | P. |
|---|---|---|---|---|
| Kindness | 0.091 | 3 | 840 | 0.965 |
| Emotional sensitivity | 0.214 | 3 | 840 | 0.887 |
| Experience openness | 0.035 | 3 | 840 | 0.991 |
| Diligence and integrity | 0.985 | 3 | 840 | 0.400 |
| Extroversion | 1.545 | 3 | 840 | 0.652 |

Note: *P<0.05

**P<0.01

**Table 8. Personality traits of different ages and multiple comparative LSD difference test.**

|  | Kindness | Emotional sensitivity | Openness to experience | Diligence and integrity | P5(Extroversion) |
|---|---|---|---|---|---|
| V1 < 30 | 2.235±1.1911 | 2.964±0.7844 | 2.758±1.0768 | 2.699±1.0426 | 2.616±1.2740 |
| V2 = 30–40 | 2.755±1.2166 | 2.817±0.8209 | 3.091±1.0908 | 3.086±1.1277 | 3.126±1.3771 |
| V3 = 41–50 | 2.516±1.2149 | 2.969±0.8095 | 2.806±1.0399 | 2.912±1.1279 | 2.898±1.3180 |
| V4 > 50 | 2.714±1.2554 | 2.797±0.9139 | 3.141±1.0761 | 3.063±1.3178 | 3.313±1.4219 |
| V1-V2 | -0.5196* | 0.1297 | -0.3334* | -0.3871* | -0.5099* |
| V1-V3 | -0.2810 | -0.0232 | -0.0481 | -0.2126 | -0.2815 |
| V1-V4 | -0.5060 | 0.1494 | -0.3826 | -0.3636 | -0.6960 |
| V2-V3 | 0.2386 | -0.1529 | 0.2853 | 0.1745 | 0.2284 |
| V2-V4 | 0.0135 | 0.0197 | -0.0492 | 0.0235 | -0.1862 |
| V3-V4 | -0.2250 | 0.1725 | -0.3345 | -0.1590 | -0.4145 |

Note:*P<0.05

**P<0.01

leisure service workers; Computer, finance and real estate; Education, scientific research and technical service workers, among the seven groups, there are significant differences in five dimensions of kindness, emotional sensitivity, experience openness, diligence and integrity and extroversion (P < 0.05). There were significant differences between the group of wholesale, retail, catering, leisure service workers and the group of education, scientific research and technical service workers in the four dimensions of kindness, experience openness, diligence and integrity and extroversion (P < 0.05). This shows that there are significant differences in personality traits among different occupational groups.

(3) Analysis based on educational background

The personality traits of different educational backgrounds are statistically analyzed by ANOVA. The homogeneity test of ANOVA is shown in Table 11 below. The variance P values of emotional sensitivity and diligence integrity are > 0.05. Accept the hypothesis of homogeneity of variance, and conduct corresponding analysis.

As shown in Table 12, further multiple comparison LSD difference test shows that there are significant differences in the two personality trait dimensions of kindness and extroversion between high school and below educated groups and college educated groups, P < 0.05; There were also significant differences between undergraduate and graduate groups in four personality traits: kindness, openness to experience, diligence and integrity and extroversion (P < 0.05); This shows that the level of education and culture is an important variable to determine the difference of personality traits in sports tourism. It is necessary to intervene these

**Table 9. Statistical table of variance homogeneity test of personality traits of different occupations.**

|  | Levene statistic | Free degree 1 | Free degree 2 | P. |
|---|---|---|---|---|
| Kindness | 0.214 | 6 | 837 | 0.887 |
| Emotional sensitivity | 1.095 | 6 | 837 | 0.366 |
| Experience openness | 0.980 | 6 | 837 | 0.398 |
| Diligence and integrity | 2.753 | 6 | 837 | 0.013 |
| Extroversion | 0.035 | 6 | 837 | 0.991 |

Note:*P<0.05

**P<0.01

Table 10. Purchase decision of different occupations and multiple comparison LSD difference test.

| | Kindness | Emotional sensitivity | Openness to experience | Diligence and integrity | Extroversion |
|---|---|---|---|---|---|
| V1 = organ civil | 1.176 ±0.3429 | 2.737±0.9140 | 1.763±0.6538 | 1.684±0.5771 | 1.590±0.6730 |
| V2 = Culture, health and sports | 2.607 ±1.0939 | 2.938±0.7563 | 3.052±1.0657 | 2.986±1.0190 | 2.667±1.1630 |
| V3 = Posts, telecommunications, electric power and transportation | 2.641 ±1.2149 | 2.918±0.8153 | 3.028±1.0954 | 3.063±1.0997 | 3.146±1.3384 |
| V4 = Construction, mining and manufacturing | 3.185 ±1.2140 | 3.052±0.8815 | 3.188±1.1961 | 3.458±1.1664 | 3.444±1.3102 |
| V5 = Wholesale, retail, catering and leisure service workers | 2.360 ±1.1948 | 2.722±0.5978 | 2.778±0.7669 | 2.654±0.8647 | 2.642±1.2016 |
| V6 = Computer, finance and real estate | 2.714 ±1.1112 | 2.842±0.8723 | 3.215±1.0067 | 3.169±1.0310 | 3.190±1.3540 |
| V7 = Education, scientific research and technical service workers | 3.143 ±1.1095 | 2.885±0.8111 | 3.365±0.8845 | 3.368±1.0081 | 3.521±1.2110 |
| V1-V2 | -1.4313* | -0.2003 | -1.2893* | -1.3024* | -1.0769* |
| V1-V3 | -1.4648* | -0.1808 | -1.2650* | -1.3787* | -1.5561* |
| V1-V4 | -2.0087* | -0.3149 | -1.4247* | -1.7746* | -1.8547* |
| V1-V5 | -1.1840* | 0.0150 | -1.0150* | -0.9706* | -1.0522* |
| V1-V6 | -1.5385* | -0.1051 | -1.4526* | -1.4855* | -1.6000* |
| V1-V7 | -1.9670* | -0.2885 | -1.6026* | -1.6838* | -1.9316* |
| V2-V3 | -0.0335 | 0.0195 | 0.243 | -0.0764 | -0.4792 |
| V2-V4 | -0.5774 | -0.1146 | -0.1354 | -0.4722 | -0.7778* |
| V2-V5 | 0.2474 | 0.2153 | 0.2743 | 0.3318 | 0.0247 |
| V2-V6 | -0.1071 | 0.0952 | -0.1633 | -0.1831 | -0.5231 |
| V2-V7 | -0.5357 | -0.0881 | -0.3133 | -0.3814 | -0.8547* |
| V3-V4 | -0.5439* | -0.1341 | -0.1597 | -0.3958 | -0.2986 |
| V3-V5 | 0.2808 | 0.1957 | 0.2500 | 0.4082 | 0.5039 |
| V3-V6 | -0.0737 | 0.0757 | -0.1876 | -0.1067 | -0.0439 |
| V3-V7 | -0.5022* | -0.1077 | -0.3376 | -0.3050 | -0.3755 |
| V4-V5 | 0.8247* | 0.3299 | 0.4097 | 0.8040* | 0.8025* |
| V4-V6 | -0.3545 | 0.2098 | 0.0279 | 0.2891 | 0.2547 |
| V4-V7 | -0.7831* | 0.0264 | -0.1779 | 0.0908 | -0.0769 |
| V5-V6 | 0.3545 | -0.1201 | -0.4376 | -0.5149* | -0.5478 |
| V5-V7 | -0.7831* | -0.3034 | -0.5876* | -0.7132* | -0.8794* |
| V6-V7 | -0.4286 | -0.1833 | -0.1500 | -0.1983 | -0.3316 |

Note: *P<0.05

**P<0.01

highly educated people with personality education and personality tourism products, so as to balance their personality level.

(4) Analysis based on individual income

The personality traits of different individual incomes are statistically analyzed by ANOVA. The homogeneity test of ANOVA is shown in Table 13 below. In addition to kindness, the variance P values of emotional sensitivity, experience openness, diligence and integrity and extroversion are all > 0.05. Accept the assumption of homogeneity of variance and carry out corresponding analysis.

**Table 11. Statistical table of variance homogeneity test of personality traits of different educational backgrounds.**

|                        | Levene statistic | Free degree 1 | Free degree 2 | P.    |
|------------------------|------------------|---------------|---------------|-------|
| Kindness               | 21.617           | 3             | 840           | 0.000 |
| Emotional sensitivity  | 0.029            | 3             | 840           | 0.993 |
| Experience openness    | 3.655            | 3             | 840           | 0.013 |
| Diligence and integrity| 2.295            | 3             | 840           | 0.078 |
| Extroversion           | 2.185            | 3             | 840           | 0.086 |

Note:*P<0.05

**P<0.01

As shown in Table 14, the further multiple comparison LSD difference test shows that between the group with an individual income of 3001–5000 yuan and the group with an individual income of more than 7001, there are significant differences in the three personality trait dimensions of experience openness, diligence and integrity and extroversion (P < 0.05); Between groups with individual income of 5001–7000 yuan and groups with individual income of more than 7001, there are significant differences in emotional sensitivity personality traits (P < 0.05). This shows that individual income is an important variable to determine the difference of personality traits in sports tourism.

## 4. Model construction of sports tourism risk perception and consumption behavior

### 4.1. Simultaneous regression model of sports tourism risk perception and consumption behavior

The simultaneous regression model integrates all explanatory variables into the regression equation to evaluate their influence on the dependent variables. [3] As shown in Table 15, firstly, determine all independent variables affecting sports tourism consumption behavior, mainly including nine variables: gender (D1), age (D2), marriage (X1), educational background (X2), family population (X3), personal income (X4), family income (X5), risk perception (X6) and personality traits (X7). Then, it can be seen that personality traits, risk perception and educational background have a high correlation with dependent variables

**Table 12. Multiple comparative analysis of LSD dimensions of personality traits with different educational backgrounds.**

|                          | Kindness    | Emotional sensitivity | Experience openness | Diligence and integrity | Extroversion |
|--------------------------|-------------|-----------------------|---------------------|-------------------------|--------------|
| C1 = high school and below | 1.53±0.63  | 2.60±0.75             | 2.31±0.72           | 2.27±0.79               | 1.78±0.73    |
| C2 = college             | 2.16±1.24   | 2.84±0.82             | 2.61±1.05           | 2.68±1.02               | 2.74±1.40    |
| C3 = undergraduate       | 2.14±1.14   | 2.94±0.81             | 2.63±1.04           | 2.57±1.07               | 2.52±1.28    |
| C4 = graduate            | 3.48±0.79   | 2.90±0.82             | 2.63±1.04           | 3.67±1.07               | 3.82±0.99    |
| C1-C2                    | -0.63*      | -0.24                 | -0.30               | -0.41                   | -0.96*       |
| C1-C3                    | -0.61*      | -0.33                 | -0.32               | -0.30                   | -0.74*       |
| C1-C4                    | -1.95*      | -0.29                 | -1.33*              | -1.40*                  | -2.04*       |
| C2-C3                    | 0.02        | -0.10                 | -0.02               | 0.12                    | 0.22         |
| C2-C4                    | -1.32*      | -0.05                 | -1.03*              | -0.99*                  | -1.08*       |
| C3-C4                    | -1.34*      | 0.05                  | -1.01*              | -1.11*                  | 0.22         |

Note: *P<0.05

**P<0.01

**Table 13. Statistical table of variance homogeneity test of personality traits of different individual incomes.**

|  | Levene statistic | Free degree 1 | Free degree 2 | P |
|---|---|---|---|---|
| Kindness | 5.148 | 3 | 840 | 0.002 |
| Emotional Sensitivity | 0.726 | 3 | 840 | 0.537 |
| Experience openness | 1.992 | 3 | 840 | 0.115 |
| Diligence and integrity | 2.257 | 3 | 840 | 0.082 |
| Extroversion | 0.774 | 3 | 840 | 0.509 |

Note: *P<0.05

**P<0.01

(purchase decision-making), and reach a significant level, while other explanatory variables are relatively low.

From further parameter estimation in Table 16, it can be seen from R2 = 0.310 that the overall model can explain 31% of the dependent variables. [4] If considering the simplicity of the model, the adjusted R2 is also 0.286, and the explanatory power is still relatively good, which means that these variable indicators and purchase decision variables can really explain risk perception. If we further examine the explanatory power of each explanatory variable, we can find that educational background (X2), personal income (X4) Risk perception (X6), personality traits (X7) the four variables have significant explanatory power. Among them, the beta coefficient of educational background is 0.168, t = 2.708, P = 0.007; the beta coefficient of personal income is 0.195, t = 2.75, P = 0.006; the beta coefficient of risk perception is 0.164, t = 2.265, P = 0.024; the beta coefficient of personality traits is 0.269, t = 3.632, P = 0.000. The explanatory power of other variables is not significant.

By summarizing the simultaneous regression model, it can be found that, as shown in Fig 1, each explanatory variable has different effects on the purchase decision of sports tourism consumption. [5] Among them, the effects of marriage, educational background, personal income, risk perception and personality traits on the purchase decision are 0.104, 0.163, 0.195, 0.164 and 0.269 respectively. The influence of these variables on the purchase decision of sports tourism consumption is positive, and the influence of other variables, including family population and family income, on the purchase decision is negative. Among other variables, personality traits have the highest influence, followed by personal income, risk perception and educational

**Table 14. Multiple comparative analysis of LSD dimensions of personality traits with different personal incomes.**

|  | Kindness | Emotional sensitivity | Experience openness | Diligence and integrity | Extroversion |
|---|---|---|---|---|---|
| E1 = < 3000 yuan | 2.58±0.60 | 2.90±0.71 | 2.94±1.06 | 3.02±1.04 | 2.90±1.31 |
| E2 = 3001–5000 yuan | 2.4±1.31 | 2.88±0.83 | 2.77±1.14 | 2.76±1.20 | 2.79±1.39 |
| E3 = 5001–7000 yuan | 2.66±1.14 | 3.03±0.79 | 3.07±1.09 | 3.00±1.09 | 2.99±1.33 |
| E4 = > 7001 yuan | 2.76±1.06 | 2.68±0.90 | 3.27±0.83 | 3.22±0.93 | 3.33±1.27 |
| E1-E2 | 0.17 | 0.02 | 0.17 | 0.25 | 0.10 |
| E1-E3 | -0.08 | -0.13 | -0.13 | 0.017 | -0.10 |
| E1-E4 | -0.18 | 0.22 | -0.33 | -0.21 | -0.44 |
| E2-E3 | -0.25 | -0.15 | -0.30 | -0.24 | -0.20 |
| E2-E4 | -0.35 | 0.20 | -0.50* | -0.46* | -0.54* |
| E3-E4 | -0.10 | 0.35* | -0.20 | -0.22 | -0.34 |

Note: *P<0.05

**P<0.01

**Table 15. Sports tourism risk perception consumption behavior related data matrix.**

| variable | average | standard deviation | D1 | D2 | X1 | X2 | X3 | X4 | X5 | X6 | X7 |
|---|---|---|---|---|---|---|---|---|---|---|---|
| D1 Gender | 0.55 | 0.499 | 1.000 | | | | | | | | |
| D2 Age | 1.95 | 0.861 | 0.288 | 1.000 | | | | | | | |
| X1 Marriage | 1.29 | 0.456 | -0.257 | -0.612 | 1.000 | | | | | | |
| X2 Educational background | 3.02 | 0.884 | -0.043 | -0.070 | -0.061 | 1.000 | | | | | |
| X3 Family population | 2.00 | 0.478 | -0.053 | 0.008 | -0.045 | -0.043 | 1.000 | | | | |
| X4 Individual income | 2.25 | 0.935 | 0.248 | 0.316 | -0.322 | 0.151 | 0.059 | 1.000 | | | |
| X5 Family income | 2.83 | 1.058 | -0.064 | 0.075 | -0.115 | 0.089 | 0.271 | 0.577 | 1.000 | | |
| X6 Risk perception | 2.9654 | 0.78857 | -0.028 | 0.055 | -0.050 | 0.429 | -0.037 | -0.001 | 0.006 | 1.000 | |
| X7 Personality traits | 2.844 | 0.9469 | -0.025 | 0.094 | -0.064 | 0.454 | -0.092 | 0.076 | 0.049 | 0.697 | 1.000 |
| Y Purchase decision | 3.002 | 0.8156 | 0.087 | 0.106 | -0.039 | 0.364 | -0.145 | 0.171 | -0.007 | 0.422 | 0.473 |

background. [6] In other words, the difference of personality traits has the greatest impact on consumers' choice of sports tourism consumer products.

Finally, considering the characteristics of the simultaneous regression model, even the explanatory variables that didn't reach the significant level cannot be ignored, they should be included in the model, so as to obtain the final equation as follows:

$$Y = 0.080D1 + 0.076D2 + 0.185X1 + 0.150X2 - 0.136X3 + 0.170X4 - 0.091X5 + 0.170X6 + 0.23X7 - 1.102$$

## 4.2. Stepwise regression models of sports tourism risk perception and consumption behavior

Stepwise regression models use stepwise regression analysis strategy to select predictive variables with explanatory power. The purpose of this model is to carry out predictive and exploratory research.

As shown in Table 17, X7 personality trait (0.473) has the highest correlation with the dependent variable (purchase decision), which was first selected into the regression equation

**Table 16. Sports tourism risk perception consumption behavior simultaneous regression model estimation results and abstract.**

| DV = Purchase decision | Non standardized coefficient | | Standardized Beta | t | P | Collinearity | |
|---|---|---|---|---|---|---|---|
| | Beta | Se | | | | Tolerance value | VIF |
| (constant) | 1.102 | 0.384 | | 2.870 | 0.040 | | |
| D1 Gender | 0.080 | 0.091 | 0.049 | 0.872 | 0.384 | 0.828 | 1.207 |
| D2 Age | 0.076 | 0.064 | 0.080 | 1.179 | 0.240 | 0.566 | 1.768 |
| X1 Marriage | 0.185 | 0.118 | 0.104 | 1.570 | 0.118 | 0.593 | 1.685 |
| X2 Educational background | 0.150 | 0.055 | 0.163 | 2.708 | 0.007 | 0.716 | 1.397 |
| X3 Family population | -0.136 | 0.092 | -0.080 | -1.490 | 0.137 | 0.897 | 1.115 |
| X4 Individual income | 0.170 | 0.062 | 0.195 | 2.752 | 0.006 | 0.512 | 1.951 |
| X5 Family income | -0.091 | 0.052 | -0.118 | -1.749 | 0.081 | 0.566 | 1.768 |
| X6 Risk perception | 0.170 | 0.075 | 0.164 | 2.265 | 0.024 | 0.491 | 2.039 |
| X7 Personality traits | 0.232 | 0.064 | 0.269 | 3.632 | 0.000 | 0.472 | 1.120 |
| Overall model | R2 = 0.310 adj R2 = 0.286 | | | | | | |
| | F (6,836) = 13.298 (P = 0.000) | | | | | | |

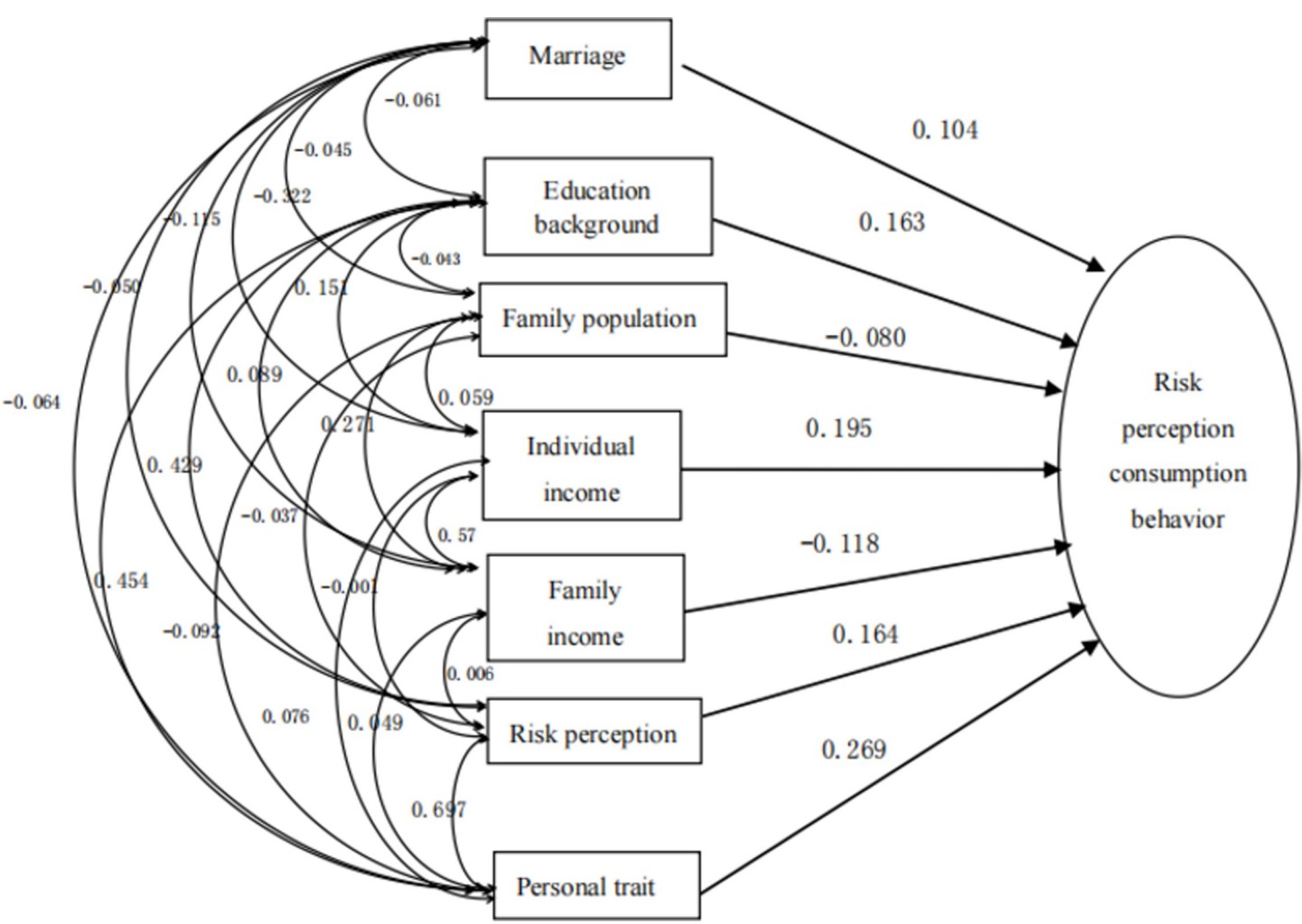

**Fig 1. Simultaneous regression model of sports tourism risk perception and consumption behavior.**

to form a single factor stepwise regression model 1, R2 = 0.473 224, the single factor of personality can explain 22.4% of people's decision-making process of sports tourism consumption. [7] Then, educational background as the second factor, because the partial correlation coefficient between it and its dependent variable is high, and the predictive power reaches the significant level of 0.05, it is selected as the second variable in the model to form a two-factor stepwise regression model 2, R2 = 0.05252, personality traits and educational background can explain 25.2% of people's decision-making process of sports tourism consumption. [7] Then, according to this, the explanatory variables of personal income, family population and risk perception enter the regression equation, and gradually form a three factor, four factor and five factor stepwise regression model of sports tourism consumption behavior, with R2 of 0.5, respectively, 265, 0.283 and 0.296, the final five factor model can explain 29.6% of people's decision-making process of sports tourism consumption [8].

Further analysis shows that, as shown in Figs 2–6, with the entry of other explanatory variables, the influence of personality traits on sports tourism consumption behavior (purchase decision) decreased from 0.473 to 0.287. [9] Similarly, the influence of educational background on sports tourism consumption behavior (purchase decision) also decreased from 0.187 to

**Table 17. Sports tourism risk perception consumption behavior stepwise regression model estimation results and abstract.**

| Variables in the model | Standard deviation | Beta | t | P |
|---|---|---|---|---|
| Model 1(R2 = 0.224) X7 Personal trait | 0.046 | 0.473 | 8.911 | 0.000 |
| Model 2(R2 = 0.252)X7 Personal trait | 0.051 | 0.388 | 6.620 | 0.000 |
| X2 Educational background | 0.054 | 0.187 | 3.196 | 0.002 |
| Model 3(R2 = 0.265)X7 Personal trait | 0.050 | 0.387 | 6.649 | 0.000 |
| X2 Educational background | 0.054 | 0.170 | 2.900 | 0.004 |
| X4 Individual income | 0.046 | 0.116 | 2.213 | 0.028 |
| Model 4(R2 = 0.283)X7 Personal trait | 0.050 | 0.388 | 6.735 | 0.000 |
| X2 Educational background | 0.054 | 0.170 | 2.928 | 0.004 |
| X4 Individual income | 0.055 | 0.210 | 3.317 | 0.001 |
| X3 Family population | 0.048 | -0.163 | -2.592 | 0.010 |
| Model 5(R2 = 0.296)X7 Personal trait | 0.063 | 0.287 | 3.901 | 0.000 |
| X2 Educational background | 0.054 | 0.146 | 2.482 | 0.014 |
| X4 Individual income | 0.055 | 0.222 | 3.516 | 0.001 |
| X3 Family population | 0.048 | -0.164 | -2.621 | 0.009 |
| X6 Risk perception | 0.075 | 0.160 | 2.207 | 0.028 |

0.146. Only the influence of personal income variables on sports tourism consumption behavior (purchase decision) increased from 0.116 to 0.222, which shows that the explanatory variables affect each other, have a covariant linear relationship, and jointly affect the occurrence process of sports tourism consumption behavior (purchase decision) [10].

Finally, from the best stepwise regression model of sports tourism risk perception in Fig 6, it can be seen that, the influences of explanatory variables on sports tourism consumption behavior (purchase decision) are varies. [11] The most influential factor is still personality traits, with an influence coefficient of 0.287, followed by personal income, family income, risk perception and educational background, with their influence coefficients of 0.222, -0.164, 0.160 and 0.146 respectively. Among them, the influence of family income on purchase decision is negative, which is different from the daily reality to some extent, further logical analysis is required. Other explanatory variables have reasonable explanations. [12] With the increase

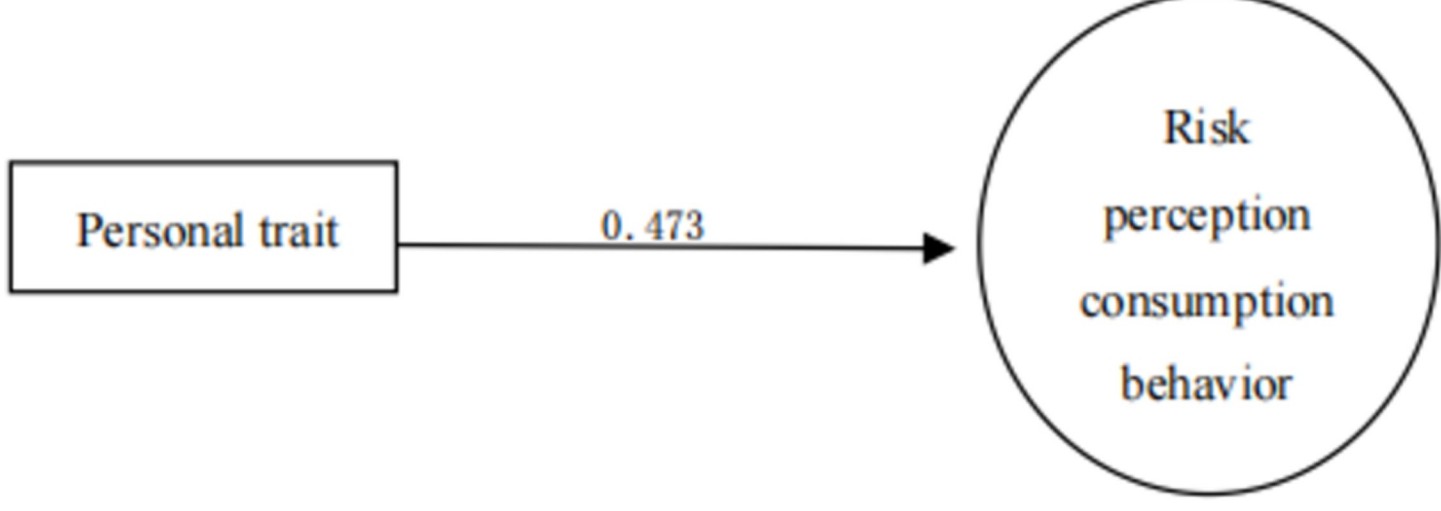

**Fig 2. Stepwise regression model of sports tourism risk perception based on personality traits 1.**

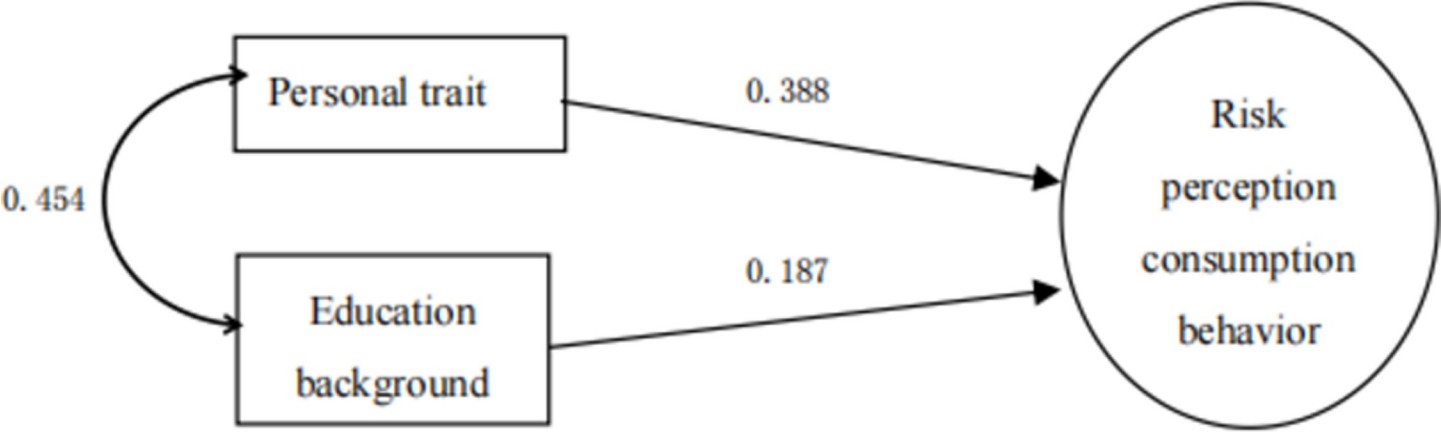

**Fig 3. Stepwise regression model of sports tourism risk perception based on personality traits and educational background 2.**

of personal income, people's willingness to buy sports tourism related products is higher; With the continuous improvement of educational background, people are more willing to buy sports tourism related products.

Finally, the whole stepwise regression equation is Y = 0.247X7 + 0.135X2 + 0.194X4– 0.126X3 + 0.166X6 + 1.323

## 4.3 Hierarchical regression model of sports tourism risk perception and consumption behavior

As shown in Table 18, The demographic variables have significant explanatory power for the dependent variable, sports tourism consumption behavior (purchase decision), R2 = 0.190, f = 9.042, P < 0.01. This shows that the demographic variable group can explain 19.0% of the variation of the dependent variable, among which the explanatory power of educational background is the strongest, (beta) reaches 0.335, and there is a significant difference, P < 0.01,

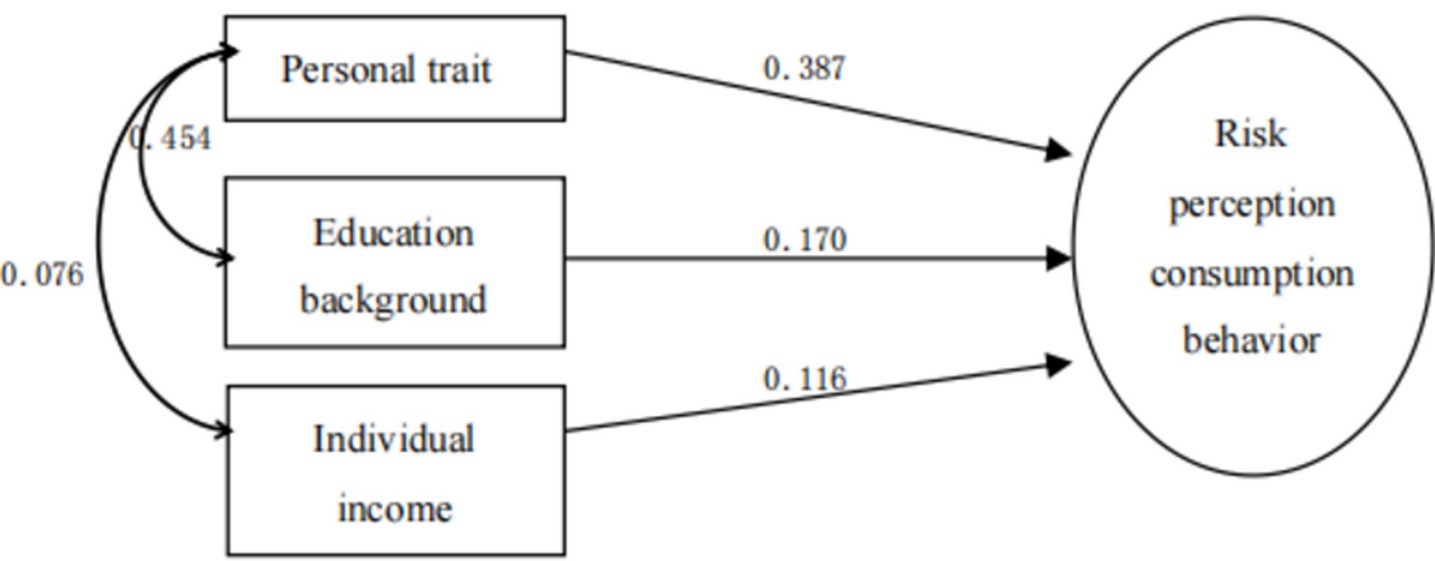

**Fig 4. Stepwise regression model of sports tourism risk perception based on personality traits, educational background and personal income 3.**

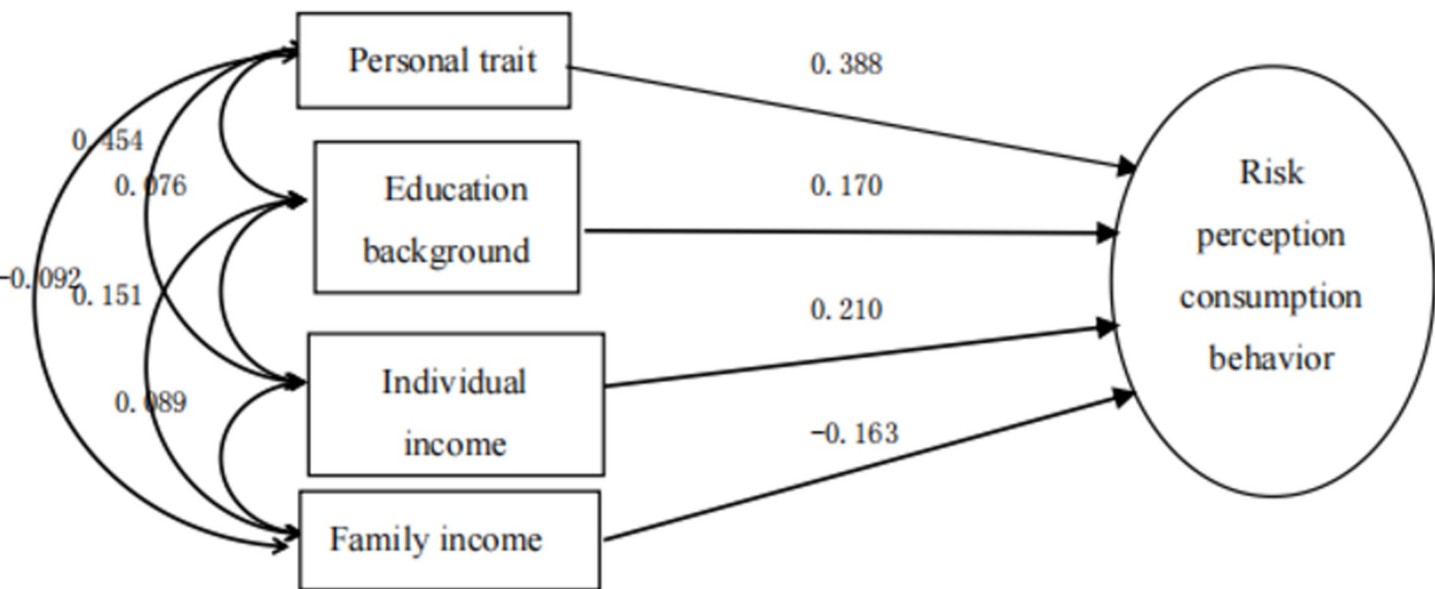

**Fig 5. Stepwise regression model of sports tourism risk perception based on personality traits, educational background, personal income and family income 4.**

followed by marriage, (beta) reached 0.218, but there was no significant difference (P > 0.05), followed by age and personal income, which reached 0.144 and 0.141 respectively, and the two variables reached significant differences (P < 0.05) [13].

After the risk perception variables of the second block were put into the model, the explanatory power for the dependent variables reached, R2 = 0.276, f = 12.783, P < 0.01. The interpretation increment Δ R2 is 0.0186, Δ f = 31.735, Δ P = 0.000 < 0.01, which shows that the investment of risk perception block can effectively improve the explanatory power of the

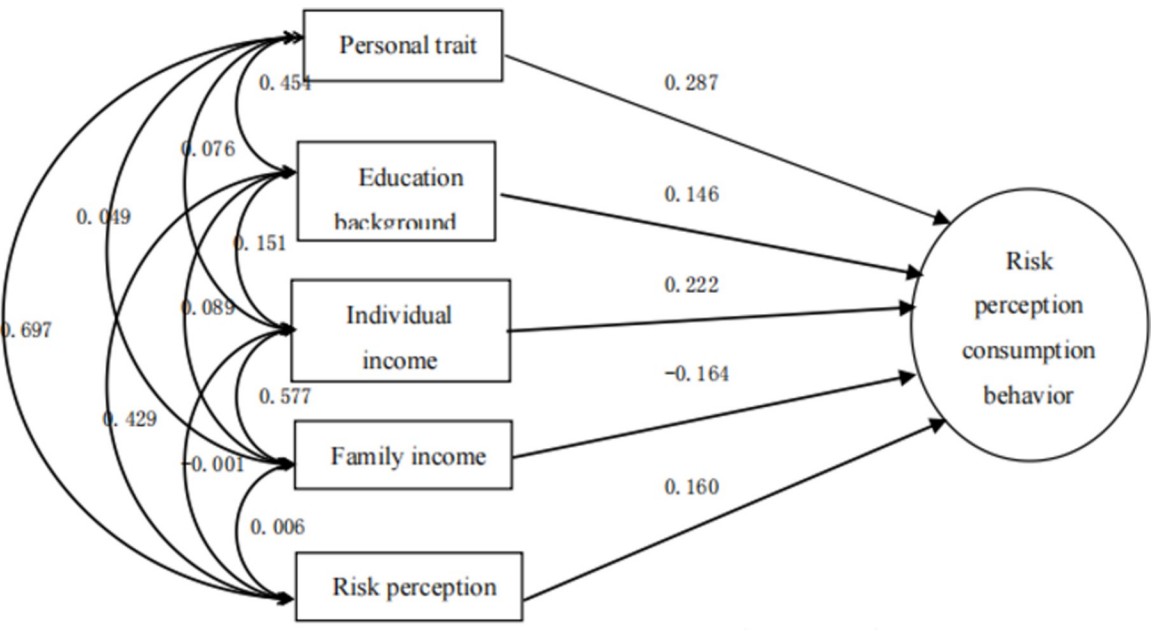

**Fig 6. The best stepwise regression model of sports tourism risk perception 5.**

**Table 18. Sports tourism risk perception consumption behavior hierarchical regression model estimation results and abstract.**

| | | Block 1 | | | Block 2 | | | Block 3 | | |
|---|---|---|---|---|---|---|---|---|---|---|
| | | Beta | t | p | Beta | t | p | Beta | t | p |
| D1 Gender | | | | 0.524 | 0.069 | | | 0.080 | 0.872 | 0.384 |
| D2 Age | | 0.063 | 0.638 | 0.036 | 0.100 | 0.737 | 0.462 | 0.076 | 1.179 | 0.240 |
| X1 Marriage | | 0.144 | 2.107 | 0.088 | 0.204 | 1.535 | 0.126 | 0.185 | 1.570 | 0.118 |
| X2Educational background | | 0.218 | 1.713 | 0.000 | 0.198 | 1.678 | 0.093 | 0.150 | 2.708 | 0.007 |
| X3 Family population | | 0.335 | 6.433 | 0.068 | -0.169 | 3.591 | 0.000 | -0.136 | -1.490 | 0.137 |
| X4 Individual income | | -0.180 | -1.832 | 0.035 | 0.175 | -1.81 | 0.070 | 0.170 | 0.006 | 0.048 |
| X5 Family income | | 0.141 | 2.118 | 0.174 | -0.085 | 2.771 | 0.006 | -0.09 | 0.081 | -0.194 |
| X6 Risk perception | | -0.077 | -1.368 | | 0.339 | -1.59 | 0.113 | 0.170 | 0.024 | 0.022 |
| X7 Personal trait | | | | | | 5.633 | 0.000 | 0.232 | 3.632 | 0.000 |
| Overall model | R2 = 0.190 | | | | R2 = 0.276 | | | R2 = 0.310 | | |
| | F = 9.042 | | | | F = 12.783 | | | F = 13.345 | | |
| | P = 0 | | | | P = 0 | | | P = 0 | | |
| | Δ R2 = 0.190 | | | | Δ R2 = 0.086 | | | Δ R2 = 0.034 | | |
| | Δ F = 9.042 | | | | Δ F = 31.735 | | | Δ F = 13.193 | | |
| | Δ P = 0 | | | | Δ P = 0 | | | Δ P = 0 | | |

model. [14] In other words, the increment of blocks is statistically significant, after controlling the influence of demographic variables, risk perception variables can contribute an additional 8% explanatory power. The explanatory power contributed by risk perception variables (beta) reached 0.339, t = 5.633, P < 0.01.

It is worth noting that the explanatory power of some demographic variables in the first block shows a downward trend, in which the changes of "marriage", "educational background" and "family population" are the most obvious, which shows that among the demographic variables controlled, the risk perception variables have a better explanatory power for purchase decision-making [15].

In the third stage, the explanatory power increment of the newly added personality trait block for the dependent variable is Δ R2, 0.034, Δ f = 13.193, Δ P = 0.000 < 0.01, which has statistical significance, which shows that the input of personality trait area group can effectively improve the explanatory power of the model, making the explanatory power of all models reach 0.310, f = 13.345, P < 0.01. Moreover, further observation of other explanatory variables shows that the beta of the explanatory variable of personality traits has decreased from 0.339 to 0.170, which shows that personality traits have a certain impact on the explanatory variables of risk perception, and there is a collinear relationship between them. After all, both explanatory variables belong to the psychological characteristics of consumers and have interactive effects [16].

According to the construction and analysis of the three blocks, we can establish three groups of different regression equations, which represents the regression model under different combinations of independent variables. After the last block enters the equation, all independent variables are included in the regression equation. At this time, the result is completely equivalent to the simultaneous regression model. Therefore, simultaneous regression is a special case of hierarchical regression model.

## 5. Discussion

The results show that, compared with women, men are more willing to buy relevant sports tourism products again, while women's willingness to buy is relatively lower, this verifies Williams et al.(2013)'s finding, so we should mobilize the initiative of female. Culture, health and

sports, government civil servants, wholesale and retail catering and leisure services, compared with the three occupational groups, construction, mining and manufacturing workers, posts and telecommunications, electric power, transportation, education, scientific research and technical services workers, computers, finance and real estate workers are more willing to buy sports tourism products, there were significant differences among some occupational groups, this is consistent with Gurtner(2014)'s findings, occupation had a certain impact on tourists' risk perception. Compared with people with high school and below, people with undergraduate and graduate education are more willing to buy relevant sports tourism products, and there are significant differences between them. It indicates that education background level would affect the risk perception of sports tourism, which support Rojo(2020)'s findings in sociodemographic characteristics of tourists.

Consumers' personality traits show significant differences in age, individual income, career and educational background, personality traits play an important role in risk perception of consumers, this further supporting the views of Cui F et al. (2016) and Zhang H et al.(2019) The simultaneous regression model of sports tourism risk perception consumption behavior shows that the four variables of educational background (X2), personal income (X4), risk perception (X6) and personality traits (X7) have significant explanatory power, and the model explanation $R^2 = 0.310$; the stepwise regression model of sports tourism risk perception consumption behavior shows that based on educational background (X2), individual income (X4), risk perception (X6), personality traits (X7) and family population (X3), the five factors model has significant explanatory power, and the model explanation $R^2 = 0.296$; through the establishment of the hierarchical regression model of consumption behavior of sports tourism risk perception, it shows that based on educational background (X2), personal income (X4) Risk perception (X6), personality traits (X7), the four variables have significant explanatory power. The model explanation $R^2 = 0.310$, which is the same as the sports tourism risk perception consumption behavior simultaneous regression model, but shows the variables that cannot be explained by the simultaneous regression model, that is, it further highlights the important role of risk perception and personality in sports tourism consumption behavior, this confirms the results of Scott & Steiger (2013) and Anuar & Sulaiman (2017) that risk perception has a direct impact on tourists' decision to travel or not.

## 6. Research limitations and suggestions

The application of risk perception in sports tourism belongs to a relatively new field, and this study is at most a preliminary exploratory study. After all, this research involves a lot of theoretical knowledge in psychology, as well as relevant theoretical knowledge in the field of economics, which is obviously cross-disciplinary and difficult. In addition, a certain amount of practical experience is needed to gain profound perceptual experience. Subsequent research will increase the accumulation of practical perceptual experience, especially supplement or revise some items through participatory survey, so as to further improve the scale. According to the research results, the following suggestions are put forward:

1. Fully expand a variety of publicity channels, further increase the publicity of sports tourism, improve people's understanding of sports tourism. At present, it is difficult to make people deeply understand the functional value of sports tourism products by simply using traditional publicity channels such as news reports and TV advertisements. Today's people need to experience before they buy, and are fully integrated into the experiential economy. Therefore, we need to develop new publicity channels, especially to develop experience as the core of publicity channels, than such as the annual sports tourism fair, to attract people to the fair to experience all kinds of sports tourism products, feel the function value of these

products, the cognition degree of experience also will get improved significantly. In addition, we should also use blog, microblog, we chat, video and SMS, which are different from the traditional newspaper, television and other new media publicity means to further promote sports tourism products. Is the key point is that tourism information from new media is basically all the tourists own body experience shows, rather than business and tourism destination of self-promotion, so new media information easier to obtain the choice of tourists and trust, the depth of understanding and an accompanying increase, naturally also prefer sports tourism products.

2. Effective market segmentation, through the establishment of sports tourism insurance system, to protect the interests of sports tourism consumers, reduce people's perception of sports tourism risk. According to the market segmentation results, it is necessary to carry out various forms of sports tourism risk perception intervention for the age group of 31–40 years old. On the one hand, it is necessary to publicize some crisis treatment methods in the process of sports tourism, on the other hand, it is necessary to let people fully understand the sports tourism insurance system, and conditionally add some insurance clauses in line with the characteristics of sports tourism, standardize the scope of sports tourism guarantee, especially the further development of sports tourism insurance products. Such as sports travel accident insurance, sports tourism transportation liability insurance, travel accident liability insurance, and so on, to ensure that sports tour visitors personal safety, of course, if conditional word, can further build sports tourism emergency rescue service center, maximum reduce their risk perception level, In this way, the sports tourism consumption market will attract more sports consumers. Reduce some of their own travel in the process of economic loss fear, so that they can better invest in sports tourism consumption.

3. Through the establishment of special service channels for sports tourism, the interests of sports tourism consumers will be further safeguarded. For income groups of more than 5,000 yuan, various forms of sports tourism risk perception intervention, especially emphasizing that in the process of purchasing and experience-related sports tourism products, there will be professionals for free guidance and consultation, such as in the process of experiencing skiing as a sports tourism product, there will be special sports tourism professionals to teach you how to get on and off the chairlift, how to wear fixed ski equipment, the function of different slopes and their safety, etc., so as to ensure the effectiveness of their products and reduce their impact on the body Concerns about functional risks and psychological risks of childcare tourism.

4. A number of sports tourism products are specially tailored for the highly educated group, to ensure the quality and service of the products, stimulate their learning enthusiasm and motivation, so that they can learn more sports professional knowledge and skills in the process of experiencing these sports tourism products. With the continuous improvement of sports tourists interest in learning and the continuous improvement of sports tourists skill level, they will recognize sports tourism products, and the accompanying risk perception will also continue to decline, and eventually become lifelong customers of sports tourism products.

5. Train special sports tourism talents and guide people to choose a variety of sports tourism products scientifically. At present, sports tourism major as a new major, the corresponding construction and development of teachers and teaching materials are lagging behind, which directly affects the quality of talent training, affect the effect of the whole professional service society. No wonder, in most scenic spots it is difficult to find sports tourism professional talents, its root is also with this. Therefore, it is necessary to further strengthen the

construction of the sports tourism course adopt combination of required and elective professional courses and the category of the social practice class, different post simulation practice lesson, and other forms to bring up the students' practical ability, strengthen the practice ability, further improve and perfect the tourism talent training system at the same time, encourage all kinds of tourism enterprises to carry out talent training in various forms and channels. For example, invite well-known educational institutions and tourism experts at home and abroad to hold various tourism training courses, lectures and academic reports in Wuqing, so as to further strengthen the education and training of sports tourism talents and practitioners. In addition, also need to set up a sports tourism skilled personnel training base, by regularly holding various travel skills contest, in order to promote development, increased competition, the selection of outstanding sports tourism talents, to master and understand some of the latest sports tourism products, improve their practical ability and innovation ability, so as to better guide people into sports tourism.

## 7. Conclusion

The simultaneous regression model of sports tourism risk perception consumption behavior shows that education background (X2), personal income (X4), risk perception (X6) and personality trait (X7) have significant explanatory power. By establishing a step-to-step regression model of sports tourism risk perception consumption behavior, the five-factor model based on educational background (X2), personal income (X4), risk perception (X6), personality traits (X7) and family population (X3) has significant explanatory power. The hierarchical regression model of sports tourism risk perception consumption behavior shows that four variables based on education background (X2), personal income (X4), risk perception (X6) and personality trait (X7) have significant explanatory power, this findigs highlighting the important role of risk perception and personality trait variables on sports tourism consumption behavior.

## Author Contributions

**Conceptualization:** Gang Li.

**Data curation:** Yan Cheng.

**Formal analysis:** Jie Cai.

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
