## [Decision Letter · Decision Letter 0]

21 Dec 2022

PONE-D-22-32637Construction of risk perception consumption behavior model of sports tourism of urban residentsPLOS ONE

Dear Jie Cai,

Thank you for submitting your manuscript to PLOS ONE. After careful consideration, we feel that it has merit but does not fully meet PLOS ONE’s publication criteria as it currently stands. Therefore, we invite you to submit a revised version of the manuscript that addresses the points raised during the review process.

We look forward to receiving your revised manuscript.

Kind regards,

Lóránt Dénes Dávid, PhD

Academic Editor

PLOS ONE

5. Thank you for stating the following in the Funding Section of your manuscript:

“This investigation work received support from Shandong Undergraduate University

Teaching Reform Research Project (M2021106), Shandong Provincial Education

Science Planning Project General project (2021YB012), Shandong University of

Finance and Economics 2021 Graduate Education Quality Course Construction

Project (Sports Industry) and Shandong University of Finance and Economics 2021

second batch of graduate courses ideological and Political demonstration course

cultivation project (Sports marketing).”

Reviewers' comments:

Reviewer's Responses to Questions

**Comments to the Author**

1. Is the manuscript technically sound, and do the data support the conclusions?

Reviewer #1: Yes

Reviewer #2: No

2. Has the statistical analysis been performed appropriately and rigorously? 

Reviewer #1: Yes

Reviewer #2: Yes

3. Have the authors made all data underlying the findings in their manuscript fully available?

Reviewer #1: Yes

Reviewer #2: Yes

4. Is the manuscript presented in an intelligible fashion and written in standard English?

Reviewer #1: Yes

Reviewer #2: No

5. Review Comments to the Author

Reviewer #1: The study deals with a current topic, both the investigated area and the topic are relevant from the point of view of the research. The structure of the study is logical. It is recommended to present the correlations between the research questions and the applied methodology. The analysis of the performed tests is more descriptive, I recommend highlighting the correlations between the results. The obtained results are novel.

Reviewer #2: The paper's title is quite complicated, hard to understand - it should be more clear.

The abstract is more or less acceptable, however, it is very short and the first sentence is very long (5,5 rows...).

The introduction is not acceptable, even not for a conference proceedings. It needs a vast extension, indicating the context and the research goals.

The literature review isn't focused well, it is rather a theoretical framework which is fine but more international sources should be reviewed and analysed in a critical and analytical way.

It is not clear why the conclusions and suggestions are in one chapter. In fact, some points here are belonging rather to the conclusions chapter.

The conclusion is too short, it is not acceptable to make it in two sentences without real content and relevance.

There is no real discussion in the paper and the description of the results is too technical, no real additions to the knowledge. The content is there but not articulated properly.

6. PLOS authors have the option to publish the peer review history of their article (what does this mean?). If published, this will include your full peer review and any attached files.

Reviewer #1: No

Reviewer #2: No

---

## [Author Response · Author response to Decision Letter 0]

13 Feb 2023

Response to Reviewers

Dear reviewers,

Thank you for your valuable suggestions on the revision of our paper. Your suggestions have guided the improvement of the paper and the thinking of future research.

According to the revision opinions, we have carefully revised the paper, and now we make the following answers to the review opinions:

Reviewer #1: 

The study deals with a current topic, both the investigated area and the topic are relevant from the point of view of the research. The structure of the study is logical. It is recommended to present the correlations between the research questions and the applied methodology. The analysis of the performed tests is more descriptive, I recommend highlighting the correlations between the results. The obtained results are novel.

Q1 present the correlations between the research questions and the applied methodology.

Answer：Add explanation in 4.1.2 line 209-218 as follows:

Multicollinearity means that the model estimation is distorted or difficult to estimate accurately due to the existence of precise correlation or high correlation between explanatory variables in the linear regression model. Generally speaking, due to the limitation of economic data, the model is improperly designed, resulting in a general correlation between explanatory variables in the design matrix.

This study investigates the relationship between factors such as gender, occupation, educational background, income and personality traits and perceived risk consumption behavior, so it is necessary to use multicollinearity analysis to prevent the model estimation distortion or difficulty to estimate accurately due to the existence of precise correlation or high correlation between dependent variables.

Q2:The analysis of the performed tests is more descriptive

Answer：The analysis of the performed tests is described in more detail, line 265, line 280-281, line 312-313, line334-335.

Reviewer #2:Q1: The paper's title is quite complicated, hard to understand - it should be more clear.

Answer：Change the title to “Study of risk perception consumption behavior of sports tourism in China”

Q2:The abstract is more or less acceptable, however, it is very short and the first sentence is very long (5,5 rows...).

Answer：The abstract has been revised and the sentence has been adjusted.

Q3:The introduction is not acceptable, even not for a conference proceedings. It needs a vast extension, indicating the context and the research goals.

Answer: Background and objectives were added to the introduction based on comments, line 38-45, line 51, line 57-60.

Q4:The literature review isn't focused well, it is rather a theoretical framework which is fine but more international sources should be reviewed and analysed in a critical and analytical way.

Answer: The literature is reviewed and analyzed in a critical and analytical way.

Q5:It is not clear why the conclusions and suggestions are in one chapter. In fact, some points here are belonging rather to the conclusions chapter.

The conclusion is too short, it is not acceptable to make it in two sentences without real content and relevance.

Answer: 8.conclusion was rewritten

Q6:There is no real discussion in the paper and the description of the results is too technical, no real additions to the knowledge. The content is there but not articulated properly.

Answer: Added more discussion and result description in the paper.

---

## [Editor Report · Decision Letter 1]

28 Feb 2023

PONE-D-22-32637R1Study of risk perception consumption behavior of sports tourism in ChinaPLOS ONE

Dear Dr. cai,

Thank you for submitting your manuscript to PLOS ONE. After careful consideration, we feel that it has merit but does not fully meet PLOS ONE’s publication criteria as it currently stands. Therefore, we invite you to submit a revised version of the manuscript that addresses the points raised during the review process.

We look forward to receiving your revised manuscript.

Kind regards,

Lóránt Dénes Dávid, PhD

Academic Editor

PLOS ONE

Additional Editor Comments (if provided):

Major revision based on reviews.
---

## [Author Response · Author response to Decision Letter 1]

7 Mar 2023

A rebuttal letter, a 'Revised Manuscript with Track Changes' and a 'Manuscript' are uploaded.

---

## [Editor Report · Decision Letter 2]

15 Mar 2023

PONE-D-22-32637R2Study of risk perception consumption behavior of sports tourism in ChinaPLOS ONE

Dear Dr. cai,

Thank you for submitting your manuscript to PLOS ONE. After careful consideration, we feel that it has merit but does not fully meet PLOS ONE’s publication criteria as it currently stands. Therefore, we invite you to submit a revised version of the manuscript that addresses the points raised during the review process. Please submit your revised manuscript by Apr 29 2023 11:59PM. If you will need more time than this to complete your revisions, please reply to this message or contact the journal office at plosone@plos.org. Please include the following items when submitting your revised manuscript:A rebuttal letter that responds to each point raised by the academic editor and reviewer(s). You should upload this letter as a separate file labeled 'Response to Reviewers'.A marked-up copy of your manuscript that highlights changes made to the original version. You should upload this as a separate file labeled 'Revised Manuscript with Track Changes'.An unmarked version of your revised paper without tracked changes. You should upload this as a separate file labeled 'Manuscript'.If applicable, we recommend that you deposit your laboratory protocols in protocols.io to enhance the reproducibility of your results. Protocols.io assigns your protocol its own identifier (DOI) so that it can be cited independently in the future. For instructions see: https://journals.plos.org/plosone/s/submission-guidelines#loc-laboratory-protocols. Additionally, PLOS ONE offers an option for publishing peer-reviewed Lab Protocol articles, which describe protocols hosted on protocols.io. Read more information on sharing protocols at https://plos.org/protocols?utm_medium=editorial-email&utm_source=authorletters&utm_campaign=protocols.

We look forward to receiving your revised manuscript.

Kind regards,

Lóránt Dénes Dávid, PhD

Academic Editor

PLOS ONE

Journal Requirements:

Additional Editor Comments:

Minor revision. The number of citations is not enough (21).

Basic papers are missing.

E.g.

http://www.efsupit.ro/images/stories/art%207.%20manuscipt_JPES_Bujdoso_David_rev.pdf

and

https://www.jocpr.com/articles/on-the-development-of-sports-tourism-in-china.pdf

---

## [Author Response · Author response to Decision Letter 2]

30 Jun 2023

Journal Requirements:

Response：The reference list is complete and correct. We did not cited papers that have been retracted.

Additional Editor Comments:

Minor revision. The number of citations is not enough (21). Basic papers are missing.

Response：12 references have been added, bringing the total to 33.

---

## [Editor Report · Decision Letter 3]

4 Jul 2023

Study of risk perception consumption behavior of sports tourism in China

PONE-D-22-32637R3

Dear Dr. Jie Cai,

We’re pleased to inform you that your manuscript has been judged scientifically suitable for publication and will be formally accepted for publication once it meets all outstanding technical requirements.

Kind regards,

Lóránt Dénes Dávid, PhD

Academic Editor

PLOS ONE

Additional Editor Comments (optional):

Accept in current form.
---

## [Editor Report · Acceptance letter]

7 Jul 2023

PONE-D-22-32637R3 

Study of risk perception consumption behavior of sports tourism in China 

Dear Dr. cai:

I'm pleased to inform you that your manuscript has been deemed suitable for publication in PLOS ONE. Congratulations! Your manuscript is now with our production department. 

Kind regards, 

on behalf of

Dr. Lóránt Dénes Dávid 

Academic Editor

PLOS ONE